# Heavy Youngsters—Habitat and Climate Factors Lead to a Significant Increase in Body Weight of Wild Boar Females

**DOI:** 10.3390/ani13050898

**Published:** 2023-03-01

**Authors:** Friederike Gethöffer, Oliver Keuling, Claudia Maistrelli, Tobias Ludwig, Ursula Siebert

**Affiliations:** Institute for Terrestrial and Aquatic Wildlife Research, University of Veterinary Medicine Hannover, Foundation, Bischofsholer Damm 15, 30173 Hannover, Germany

**Keywords:** *Sus scrofa*, reproduction, weight increase, habitat, climate factors

## Abstract

**Simple Summary:**

Long-term monitoring of wild boar reproduction in Lower Saxony, Germany, facilitated the description of body weight development in two different habitat types over an 18-year period. Here, not only the sampling year but also habitat and climate factors proved to be of considerable importance for the development of *Sus scrofa* body weight. Additionally, of particular interest were differences in the timing of puberty achievement among habitats.

**Abstract:**

As one of the most abundant game species in Europe, European wild boar (*Sus scrofa*) populations prove highly adaptable to cultivated landscapes. The ongoing process of climate change and the high agricultural yields seem to further optimize the living conditions for this species. In long-term reproduction monitoring, we collected data on the body weight of wild boar females. Over an 18-year period, the body weight of wild boar females increased continuously, then stopped and decreased. It was possible to detect differences between the body weights of animals from forest and agricultural areas. For these areas, differences in body weight development also led to a significant distinction in the onset of puberty. We conclude that, even in a highly cultivated landscape, forested areas provide habitat characteristics that may strongly influence reproduction. Second, with dominant agricultural areas in Germany, wild boar reproduction has been favored in recent decades.

## 1. Introduction

The wild boar (*Sus scrofa*) is a popular and widely abundant game species [1], and populations have increased throughout Europe in recent decades [2,3,4,5,6]. It is an opportunistic omnivore with a high adaptability to changing environmental conditions [7,8,9,10,11]. The wild boar has by far the highest reproductive potential and fecundity of all ungulate species worldwide in relation to its body mass [12,13,14,15,16]. Combined with a low natural adult mortality rate in temperate climates [2], it can achieve high population densities in a very short time period [17]. 

Due to its increasing population, it has become a very important species from an economical point of view [1,18]. It can transmit diseases (like the African swine fever) to livestock [19,20,21] and cause damage to agriculture [17,22,23]. Several aspects that contribute to wild boar population growth have been studied and discussed. Among them, the most important factors are a reproductive success or lower juvenile mortality due to optimal feeding [24,25] and mild climatic conditions [26,27,28].

Several parameters of wild boar reproductive biology, such litter size [16,29], sex ratio and parturition time (unpublished own data) are positively related to maternal body weight. A body weight increase in free-living wild boar populations was reported from different European habitats [30,31], arising the question if similar developments could be observed from study areas situated in parts of Germany that were dominated by a high extent of cultivated landscape. 

Therefore, the aim of this research was to investigate whether the body weight of female wild boars in some areas of northern Germany increased over an 18-year period and whether the increase in body weight was different among age classes and habitats. Here, the influence of external (e.g., climate and mast) and internal (e.g., age and maturity) factors on body weight was tested, and, in addition we also tested for a possible effect on the onset of puberty. We hypothesized that body weight increased in all age classes over the years and that comparisons between forest and agricultural habitats showed a greater weight increase in the latter. 

## 2. Materials and Methods

### 2.1. Study Area 

The study area was situated in the East of the federal state of Lower Saxony (northern Germany) (52.36° N, 10.35° E) and comprised about 5500 km^2^ with altitudes ranging from 60 to 130 m asl [32]. Based on data from the German Meteorological Service, the average annual temperature since 2000 was 9.7 °C, and the average annual precipitation was about 800 mm. The trend of mean temperature is increasing, and mean precipitation is decreasing [33].

The study area comprised either forestal–agricultural habitats (fah) or represented a large contiguous forestal habitat (fh, Figure 1). Within the total area, one large block of forest (430 km^2^) was interpreted as a separate territory, containing only 3% of agricultural areas, and thus, agricultural food was available for wild boars only to a limited amount. Forest composition was almost the same as in the total area, while the forested area additionally comprised 2% of heathland and 5% of settlement and infrastructure. This area could be interpreted as forestal habitat (fh, Figure 1), while the agriculturally dominated part of the study area was about 5500 km^2^, with 61% auf agricultural areas and 22% of forest. The remaining 17% were buildings and infrastructure. 

The agricultural land was characterized by 80% of energy-enriched products (e.g., grain, sugar beets, oilseed, rape, maize, leguminoses and potatoes) in 2020. The remaining 20% consisted of heathland, fallow land and horticultural products [34]. Since 2007, only slight changes within the different field crops were reported [34]. The forest consisted of 18.9% oak (*Quercus* spp.), 11.2% beech (*Fagus sylvatica*), 7.6% other deciduous trees (e.g., *Betula* spp, *Acer* spp., *Tilia* spp., *Sorbus aucuparia*), 45.5% pine (*Pinus* spp.) and 16.6% other coniferous trees (*Larix* spp., *Picea abies, Pseudotsuga menziesii*).

### 2.2. Data Collection

Data were collected from 3517 free-ranging hunted wild boar females in the study area during the hunting years from 2003 until 2020. Each hunting year (indicated as “year” in the following text) started in April and ended in March. Since the majority of samples derived from November–January, we focused our analyses on these months of the hunting year (*n* = 3.367, Table 1). The age of the wild boar females was identified by tooth determination [35]. Following the age identification, female wild boars were grouped into three classes (*n* = 2024 piglets: ≤12 months; *n* = 866 yearlings: ≥13–24 months; *n* = 477 adults: ≥25 months, Table 1). Their dressed body weight (without digestive system, heart, lungs, liver, reproductive tract, and blood, hereafter “bw”) was recorded. Dissection and examination of the reproductive tract (uterus and ovaries) was performed according to our own protocols [36]. Consequently, the onset of sexual maturity was assumed when ovaries had follicles ≥0.4 cm in diameter.

### 2.3. Statistical Analysis

We described with Tukey’s Honest Significance Difference (HSD) test differences between mean bw in the three age classes both overall and within the two habitat types. Then, we used generalized linear models (GLM, [37]) with identity link to analyze the development of bw over the hunting years, including age class, habitat type, tree crop and weather variables (Table 2). Explanatory variables were checked for correlations among each other [38], but multicollinearity did not affect our analyses, as checked by correlations and variance inflation. Testing for both linear and unimodal relationships with the dependent variable (bw), we used the AIC (Akaike Information Criterion) from univariate models with each numerical weather variable to decide about their inclusion in the GLM. We stopped when adding a new variable did not result in a lower AIC value. We also included the hunting index as a measure of hunting intensity for each hunting year. We used a GLM with logit link and binomial error structure to find predictors of wild boar puberty. All analyses were performed using R software version 4.1.0 [39].

## 3. Results

### 3.1. Overall Body Weight

Tukey’s HSD test revealed highly significant differences between age classes in the whole sample. Mean bw for piglets, yearlings, and adults was 27.6 kg [sd 8.6, *n* = 2024], 54.0 kg [sd 10.5, *n* = 866] and 66.7 kg [sd 12.9, *n* = 477], respectively.

In all age classes, we found an increase in bw that resulted in significant differences between some months, following a seasonal pattern during autumn and winter. A bw difference (diff) between November and January was significant in yearlings (Tukey’s HSD test: *p* = 0.05, diff = −2.3 kg) and adults (Tukey’s HSD test: *p* < 0.01, diff = −4.2 kg). In piglets, there was a significant bw difference between November and December (Tukey’s HSD Test: *p* < 0.0001, diff = 3.1 kg) and, like in the other two age classes, between November and January (Tukey’s HSD Test: *p* < 0.0001, diff = −3.9 kg). The weight increase in piglets from November to December was especially pronounced in the fh (diff = 5.0 kg) and, to a lesser extent, in the fah (diff = 2.3 kg).

### 3.2. Effect of Habitat on Body Weight

Mean bw was higher in fah compared to fh habitats (Figure 2). The difference of 2.8 kg was significant (Tukey’s HSD *p* > 0.01). Differences between habitat types were particularly pronounced in yearlings. In this age class, the mean bw of animals from the agricultural habitat was 8.0 kg higher than that from forest habitat (Tukey’s HSD *p* > 0.01). In piglets and adults, the weight difference between habitats was 5.2 kg and 5.5 kg, respectively.

### 3.3. Temporal Trends in Body Weight and External Factors

Bw differed per age class and habitat, but the overall trend was similar in both factor variables. In all models, the relationship of bw with hunting year was best described by a fourth-order polynomial that visually suggested a slight increase until about 2017, followed by a decrease until 2020 (Figure 2). Fluctuations in bw were more pronounced for wild boar in fh.

Strong beech mast benefited especially adult animals in fh (Figure 3), while oak mast did result in differences in bw without any significance. Among all age classes, climate factors such as frost periods in February negatively affected bw, while higher precipitation in May and July had a positive impact (Figure 4). The inclusion of the hunting index further improved the final model, with the bw being negatively correlated with this predictor of hunting pressure (Figure 4), which was again true for all age classes. 

### 3.4. Predictors of Wild Boar Puberty

In our data, piglets reached puberty approximately at a bw of 30 kg, and in fh, this was achieved at an age of 11 months, whereby piglets of fah seemed to achieve the crucial body weight at a lower age, i.e., about 8–9 months (Figure 5). As a result, in fah, 80 percent of the 8-month-old piglets had already reached puberty, while in forest areas, only 64 percent did. In years with beech masts, the probability of puberty was slightly higher than in years without mast (Figure 6).

## 4. Discussion

During the last years, a permanent weight increase in female wild boars of all age classes in Lower Saxony culminated in 2017 and decreased from thereon for unknown reasons, therefore, partly supporting our hypothesis on a permanent weight increase of the species in the wild. The unexpected halt and decrease in body weight might be in line with extremely arid summers occurring since 2018 in the region [33]. By highlighting a significant distinction of body weight between animals from forestal (fh) vs. agricultural (fah) habitats, our second hypothesis proved right. In Poland, a similar increase of wild boar body weight has been reported, and the distance to forestal areas also turned out to be a significant factor [30]. Interestingly, the difference in body weight has an additional impact on puberty achievement, and here, mast seeding events significantly improve the condition of animals situated in forestal habitats. This is supporting our results from earlier studies, indicating that the occurrence of splendid masts of oak or beech result in a higher reproductive outcome, especially in forestal areas [29]. Additionally, the precocious puberty of female and male [40] wild boars in our study area supports the theory of a high share of piglets in reproduction. Anyway, the proven distinction of habitats longs for discussion.

Various studies reflect that wild boars’ spatial behavior is highly adaptive and diverse [41,42,43], especially among different European countries [44,45,46,47,48]. In our study, we focused on adjacent areas of agricultural habitats and woodland, where the woodland was characterized by its consistency. We assume that wild boar populations inhabiting agricultural areas near the edge of the forest habitat do enter it and vice versa but that the majority of the animals inhabiting the pure forest area is concentrating on the forest, which is also providing enough forage, especially under climate change conditions [49,50]. This is supported by the fact that puberty is reached by piglets inhabiting agricultural dominated areas at lower ages, while mast occurrence influences the onset of puberty, especially among forest-inhabiting animals. While the hybridization of domestic pigs and wild boars might lead to an adjustment in reproductive traits in the latter, there is no indication that hybridization might be responsible for the habitat-dependent differences we found in this study. In Europe, hybridization mainly occurred during the outside feeding of domestic pigs in the Roman Empire and Middle Ages [51], and in some countries, it was also observed after World War II or even recently [52,53]. A wide variation in the degree of ongoing hybridization in European countries is stated, referring to the type of pig farming as one explaining factor [54]. In this context, Germany is one of the countries where industrial pig farming is most common [55] and, therefore, ongoing hybridization is less likely. Nevertheless, the differences in the reproductive performance of wild boars between Northern and Southwestern Germany that were found in former studies were not tested for differences in hybridization with domestic pigs so far [32,35], and studies that precisely perform German wild boar genome analyses would certainly be helpful to interpret reproduction in this species more accurately. While both sexes of wild boars seem to reach puberty on highly weight-dependent conditions [40] and have gained weight continuously over the last years, it was surprising to observe that mast seeding events in forestal habitats appear to additionally influence the onset of puberty. The meaning of mast seeding events for wild boar reproduction is controversially discussed, and although, e.g., wild boars in France also seem to respond to mast seeding events positively through a higher breeding proportion, the effect seems to be caused mainly by their weight increase [56]. This could be due to the difference in habitats, with the French Mediterranean forest seemingly serving as a main feeding source for wild boars [56], while our study area has been dominated by agricultural land use from the beginning of the monitoring. The high adaptability of wild boars might lead to aligned feeding strategies, in which the occurrence of mast seeding events might not provoke an imminent increase of body weight but resolve in a short-time adjustment of body fat and fatty acids, discussed as “flush feeding” in pigs [57], that also seems to have different outcomes depending on the timing and reproductive status of the animals [58,59]. Few large contiguous forestal habitats are situated in Northern Germany, and with climate change, the persistence of comparable forests is threatened [60]. Whether additional reproductive parameters, such as the time of farrowing and litter size of wild boars, will also prove to be habitat-dependent in our study areas will, therefore, be part of our future research. 

An additional impact could be shown for various external factors, such as precipitation in June and frost periods in February, leading to diminished body weights, whereas higher temperatures in March and April, as well as precipitation in May, were improving body weight. It seems reasonable that cold periods in February, with possibly farrowing females or piglets among the rut, lead to a reduced body weight of a high percentage of animals. Potentially, animals will search for shelter and be less active in foraging [61]. Likewise, it can be assumed that strong precipitation in June may lead to crop failure or at least poor harvests as a consequence and thus also results in a decrease in body weight. The meaning of seasonality in wild living species is an important topic in wildlife research [62], and in the wild boar, questions about reproductive seasonality still remain unsolved [63]. Additionally, little is known for weight development of wild boars in other European habitats, and further research can be helpful to evaluate the importance of these factors in Europe’s future climate. 

## 5. Conclusions

Within a long-term study on the reproduction of female wild boars, we were able to gain outstanding results on basic features such as body weight development and received valuable insights on the impacts of external and internal factors on puberty. We, therefore, explicitly propose to establish long-term monitoring of wildlife species to a far greater extent.

## Figures and Tables

**Figure 1 animals-13-00898-f001:**
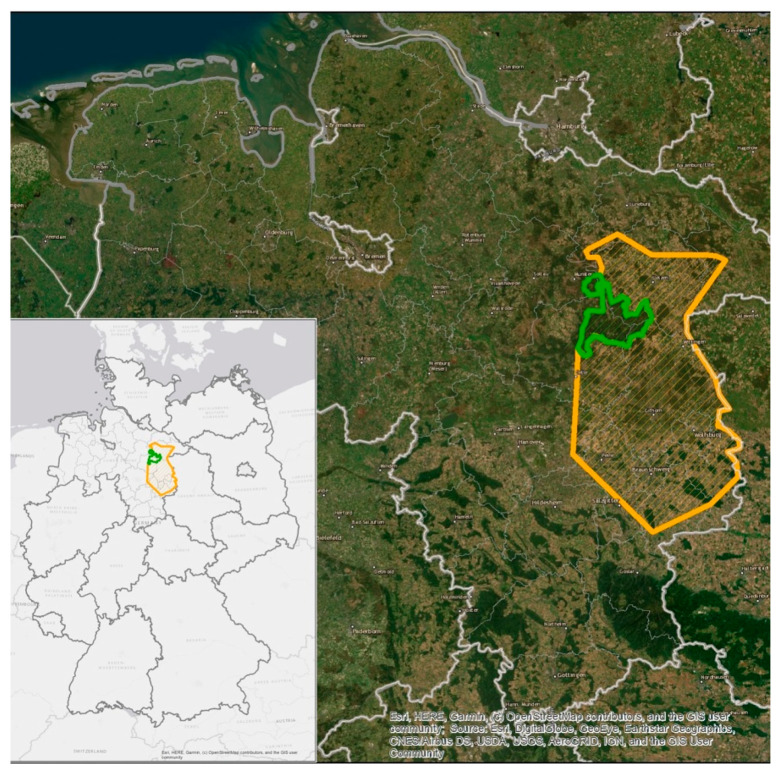
Map of Germany (**left**), indicating the site of the study area in Lower Saxony, Germany (**right** part of the figure), showing forestal–agricultural habitat (fah, outlined in orange) around the large contiguous forestal habitat (fh, outlined in green) where wild boar females of this study were sampled.

**Figure 2 animals-13-00898-f002:**
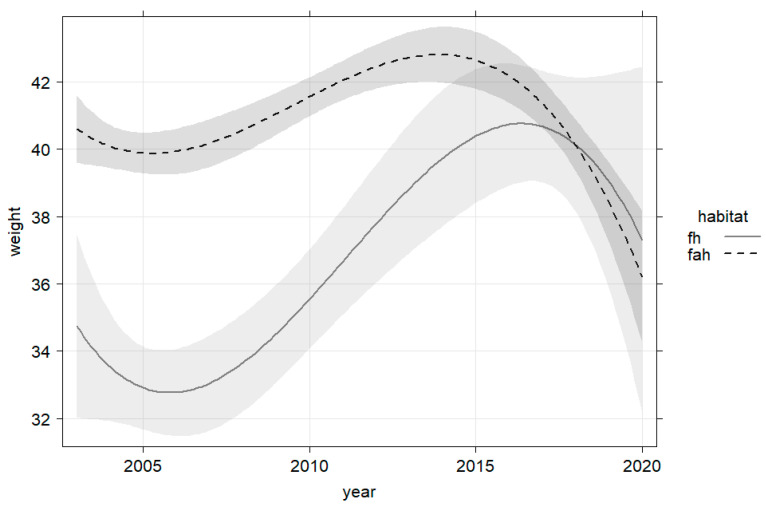
Bw (weight) of wild boar females (all ages) increased with hunting year until about 2017, followed by a decline that was less pronounced in agricultural dominated habitat fah (*n* = 2837).

**Figure 3 animals-13-00898-f003:**
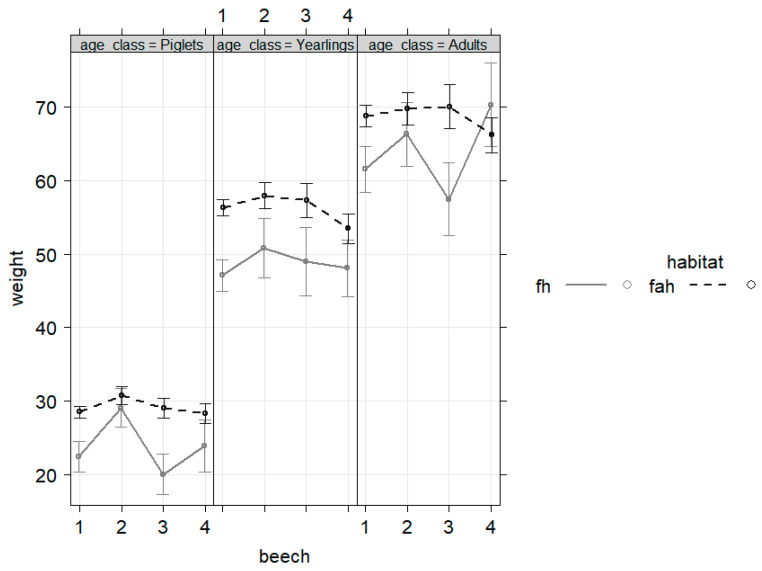
Strong beech mast (numbers represent levels according to categorization from forestry commission office Oerrel, Lower Saxony; 1 = no, 2 = part, 3 = half, 4 = full mast) resulted in (significantly) positive effects on bw only for adult wild boar females in fh (*n* = 680, *p* = 2.77 × 10^−7^ for beech = 4, age = adult, habitat = fh).

**Figure 4 animals-13-00898-f004:**
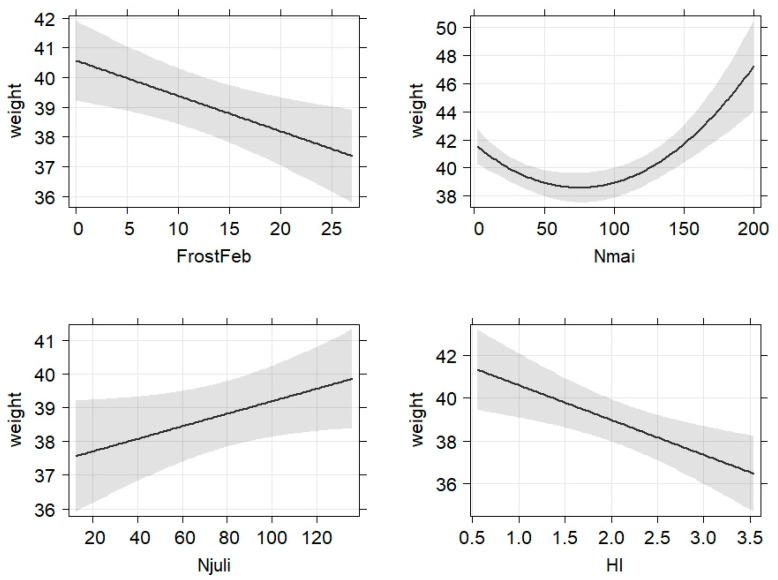
Bw (weight) of wild boar females of all ages was negatively influenced by days of frost in February (FrostFeb), lower precipitation in May (Nmai, precipitation in mm) and July (Njuli, precipitation in mm) and strong hunting pressure (HI, ratio of hunting bag/hunting area).

**Figure 5 animals-13-00898-f005:**
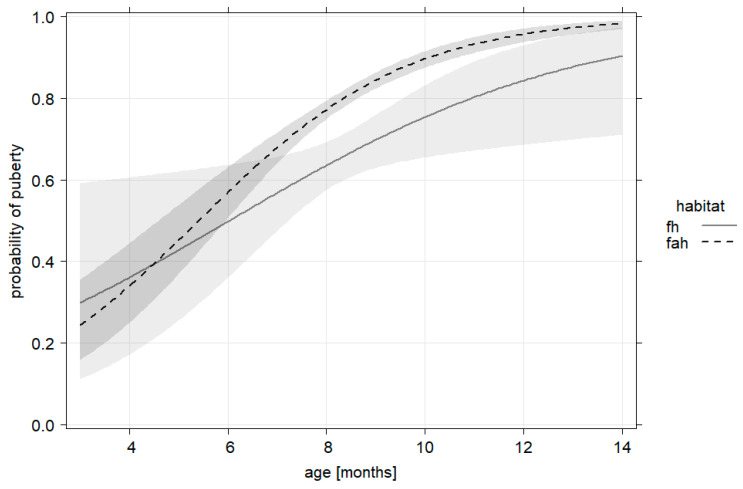
Different probabilities of puberty in the two landscape types (forest = fh, agriculture = fah), as described by age of piglets.

**Figure 6 animals-13-00898-f006:**
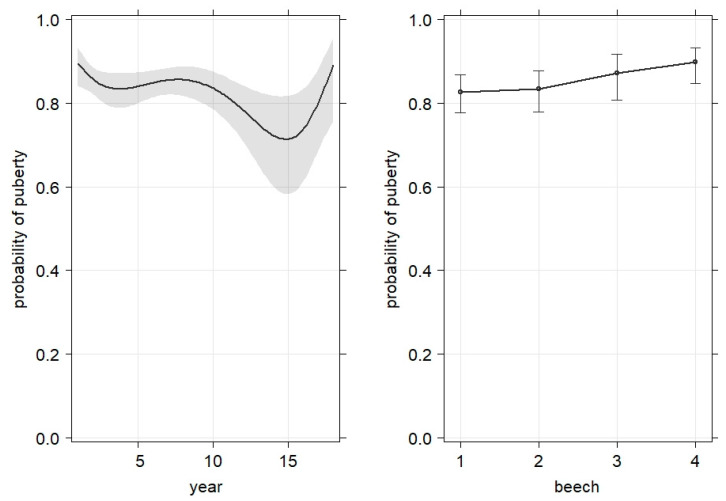
Probability of puberty slightly decreased with increasing overall bw over years (**left**) and slightly increased with beech mast (**right**; numbers represent levels according to categorization from forestry commission office Oerrel, Lower Saxony; 1 = no, 2 = part, 3 = half, 4 = full mast).

**Table 1 animals-13-00898-t001:** Distribution of wild boar females according to age and habitat.

Habitat	Piglets	Yearlings	Adults	Habitat Total
fh	306	184	103	593
fah	1718	682	374	2.774
total	2024	866	477	3.367

**Table 2 animals-13-00898-t002:** Predictors of wild boar body weight used in this study.

Variable	Measure/Explanation
Year	numerical (2003–2020)
Age class	three nominal levels: piglets, yearlings, adults
Habitat	two nominal levels: forested (fh), agricultural (fah)
Oak crop	four ordinal levels: (1–4)
Beech crop	four ordinal levels: (1–4)
Precipitation	monthly sum in mm
Temperature	monthly average (°C)
Hunting Index	wild boar annually shot per km^2^

## Data Availability

Data for land use and climate are available online via https://www1.nls.niedersachsen.de/statistik and https://www.dwd.de/DE/leistungen/ (accessed on 1 February 2023).

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
