# Peer review of "Heavy Youngsters—Habitat and Climate Factors Lead to a Significant Increase in Body Weight of Wild Boar Females"

_animals, 2023, doi:10.3390/ani13050898_

Round 1

Reviewer 1 Report

With great interest, he immediately threw himself into revising the manuscript, because according to the abstract I expected a very interesting work. After reading the manuscript, I realized that my expectations were met. Many teams around the world (especially across European countries) are working on the reproduction of wild pigs and the factors that influence reproduction. However, no one other than a team of researchers from Germany has such a long time series in the context of an objective assessment of the influence of factors (external and internal). These authors are the "parents" of methodological approaches to the evaluation of reproduction that are used by the rest of the world. Therefore, a similar study with a long time series and a larger number of samples than in this study cannot be expected in the near future.

The manuscript is very well written, appropriately structured, and the methods used correspond to the principles of correct evaluation. The Introduction contains a brief but complete list of relevant literary sources, and the Discussion contains both a comparison of one's own findings with the findings of other authors, but also an elaboration of the findings on one's own ideas, which have not yet been proven. I found no errors in the text (except for the remaining end of the parenthesis on line 13; and the need to reformulate key words that duplicate words in the title of the manuscript - on line 24) substantive or methodological. I do not feel competent to assess the level of English.

Reviewer 2 Report

The manuscript titled "Heavy youngsters - habitat and climate factors lead to a significant increase in body weight of wild boar females" interprets the body mass variations of a large wild boar population, with ecological and physiological factors useful for interpreting the demographic variations of this species.  The work is very interesting, useful in managerial terms and modern, unfortunately it is based on a not very recent literature and some considerations should be better explained to avoid being too speculative.   In detail:

- Line 28. Add the more recent reference about wild boar expansion and related emergency.   E.g.:

Tack, J. Wild boar (Sus scrofa) populations in Europe. A scientific review of population trends and implications for management. Eur. Landowners’ Organ. Bruss. 2018, 56, 29–30.

Lewis, J.S.; Corn, J.L.; Mayer, J.J.; Jordan, T.R.; Farnsworth, M.L.; Burdett, C.L.; VerCauteren, K.C.; Sweeney, S.J.; Miller, R.S. Historical, current, and potential population size estimates of invasive wild pigs (Sus scrofa) in the United States. Biol. Invasions 2019, 21, 2373–2384. https://doi.org/10.1007/s10530-019-01983-1.

- Line 29 add the reference “Petrelli, et al., Population genomic, olfactory, dietary, and gut microbiota analyses demonstrate the unique evolutionary trajectory of feral pigs. Molecular Ecology. 00:1–18.https://doi.org/10.1111/mec.16238" which determines the diet of the wild boar with molecular methods.

- Line 33. Fecundity also depends on possible hybridisation phenomena with the pig, a problem that also re-emerges when the authors discuss the anticipated puberty in wild boars that frequent the agricultural environment, here and in the discussion I would add references which addresses this issue.  

- Line 69, I would put the figure showing the study area in the main text rather than in the supplementary materials.  This also applies to figure S2 which is important in the main text.   - Lines 109 - 110, it would be correct to insert also variance accompanying the mean values.  

- Lines 121 - 126. These results refer to the graph in figure 1 which does not show the increase in BW in young people but overall data for the species.  

- Line 133 - 137 it is not clear who is the subject? If necessary insert a new figure, for example the 2a  

- Line 140 - 144. It is not clear whether this datum (Piglets reached puberty approximately at a bw of 30 kg …) derives from the data or is information from the literature. The authors should clarify this.  

- Figure 2. The authors should be report values of significance or insert a symbol (i.e. an asterisk) to indicate statistically significant comparisons  

- Figure 3. The legend should be revised. Bw (weight) of who?  

-  Line 182. The adaptive and therefore expansive capacity of wild boars depends on numerous factors, among which the sense of smell that is a trait playing a crucial key role, sometime also interpreted as pre-adaptation. It might be useful to mention Fulgione et al., 2017, Pre-birth sense of smell in the wild boar: the ontogeny of the olfactory mucosa. Zoology. https://doi.org/10.1016/j.zool.2017.05.003", dealing with this.  

- Line 187 - 190. See comment about the possible pig-boar hybridization.
